# Effects of far infrared therapy in hemodialysis arterio-venous fistula maturation: A meta-analysis

**Chiu-Feng Wu[1]⦿, Tzu-Pei Yeh[iD][2]⦿, Tzu-Chen Lin[1], Po-Hsiang Huang[1], Pin-Jui Huang[iD][3]***

**1** Department of Nursing, Ditmanson Medical Foundation Chia-Yi Christian Hospital, Chia-Yi, Taiwan,
**2** School of Nursing, China Medical University Hospital, Taichung, Taiwan, **3** Divisions of Urology, Department of Surgery, Ditmanson Medical Foundation Chia-Yi Christian Hospital, Chia-Yi, Taiwan

⦿ These authors contributed equally to this work.
* w4d5j0x@hotmail.com

## Abstract

### Introduction

Hemodialysis patients rely on stable vascular access to perform effective hemodialysis and reach good dialysis quality. However, an obstructed or under-matured arteriovenous fistula (AVF) may increase infection rate and mortality in hemodialysis patients. Far infrared (FIR) therapy might help to promote AVF maturation and reduce obstruction rate. Therefore, this meta-analysis was conducted to evaluate the effect of FIR therapy on AVF obstruction rate and maturation.

### Material and method

PubMed, Embase, the Cochrane Library, and other databases which provide publications in randomized controlled trials (RCTs) of FIR to improve AVF in patients with CKD (Chronic Kidney Disease) or HD (hemodialysis) were used to collect articles which published before February 2023. Two authors selected relevant articles independently based on pre-defined inclusion and exclusion criteria, and assessed the quality of the articles by using the Cochrane Handbook before performing a meta-analysis in Review Manager (RevMan) 5.4 software.

### Results

Four RCTs with 475 patients were included. The results of the meta-analysis showed that the FIR therapy groups had better physiological maturation at 3 months (RR = 1.22; 95% CI = 1.07 to 1.39; p = .002) and clinical maturation at 12 months (RR = 1.35; 95% CI = 1.14 to 1.60; p < .001) than the control groups without FIR therapy. The obstruction rates within 12 months were much lower in the FIR therapy groups than in the control groups (RR = 0.24; 95% CI = 0.08 to 0.68; p = .007), also, there was no statistical heterogeneity.

### Conclusions

FIR could promote fistula maturation and reduce the incidence of AVF obstruction.

**Data Availability Statement:** All relevant data are within the manuscript.

**Funding:** This research was funded by China Medical University, CMU108-TC-01, Tzu-Pei Yeh was the principal investigator. The funders had no

role in study design, data collection and analysis, decision to publish, or preparation of the manuscript.

**Competing interests:** The authors have declared that no competing interests exist.

## Introduction

According to the United States Renal Data System (USRDS) 2022 annual report, the number of people receiving hemodialysis (HD) in the United States almost doubled from 2000 to 2020, reached 480,516. In addition, the total number of people had new registrations with End-Stage Renal Disease (ESRD) in 2020 was 130,522; 109,107 patients of this population (83.9%) chose HD as renal replacement therapy [1]. To obtain the optimal HD treatment results in the clinic, patients need to rely on full-functional and stable vascular access for hemodialysis. Studies have shown that if a patient starts HD with a catheter at the beginning of dialysis, the mortality rate within 18 months is more than twice than those who with an arteriovenous fistula (AVF) at the beginning [1]. Therefore, the first-choice of access for HD is AVF, because it has fewer complications, better overall patency, and lower mortality than other types of vascular access [1–4]. However, it takes several months for a new AVF to be mature for use and the cumulative incidence of primary unassisted patency is only 59.1% [1, 5]. Approximately 50% patients had poor fistula maturity and needed endovascular or surgical procedures to promote AVF maturation [6, 7]. Despite the optimization in the maintenance of the AVF, obstruction still occurs [7, 8] or its maturation is insufficient; this may force patients to depend on the central venous catheter for longer time and increase the risks of catheter infection [6, 7]. This may make a significant impact on patients' lives and require huge medical costs and resources.

Far infrared (FIR) is an invisible electromagnetic wave. FIR therapy (FIR) uses low-power electromagnetic waves emitted from far infrared to improve human physiological functions (wavelengths of 3–100 μm). Previous studies showed that FIR therapy could effectively suppress inflammation; and it has been widely used in the clinical treatment of various diseases in recent years, such as post-catheterization patients in FIR saunas [9, 10]. The use of FIR therapy to improve vascular access flow and patency is based on the influences of local vasodilation which not only induced by thermal effects, but more importantly by non-thermal effects to improve endothelial function [11–13]. The treatment effects above may due to the ability of FIR could effectively activate the heme oxygenase-1 gene (HO-1), as HO-1 is effective in reducing inflammation. HO-1 also work on inhibiting vascular smooth muscle cell proliferation and stimulates endothelial cell regeneration at the site of endothelial injury; thereby the occurrence of vascular stenosis could be reduced [14, 15].

FIR therapy has been used in recent decade to promote AVF maturation with the goal of increasing vascular access flow and patency. In 2014, Bashar et al. [16] conducted a meta-analysis of the effects of FIR therapy on increasing the patency of primary and secondary AVF. In 2017, Wan et al. [12] also conducted a meta-analysis on the effect of FIR treatment on AVF patency, but the results of this review were too heterogeneous and most of the studies which met the inclusion criteria were not available in full text; consequently the reliability of the results was reduced. Therefore, our research team conducted a systematic review (SR) and meta-analysis (MA) to evaluate the impact of FIR therapy on AVF maturity and patency.

## Material and method

This systematic review and meta-analysis was conducted following the PRISMA guidelines for Systematic Reviews and Meta-Analyses [17].

### Search strategies

PubMed, EMBASE, Cochrane Library, Web of Science, and other databases were searched to obtain RCTs which may answer the question until February 2023. The following keywords were used and searched by using Boolean algebra: "chronic kidney disease or dialysis, hemodialysis, end-stage renal disease, renal failure, fistula, arterial fistula, graft, far-infrared therapy,

far-infrared. The bibliographies of included studies were also been searched to find out more related studies.

## The inclusion criteria were

1. Study type: Studies designed with RCTs

2. Participant Type: patients diagnosed with CKD or ESRD, and receiving regular HD treatment with AVF.

3. Intervention type: FIR therapy versus non- FIR therapy.

4. Outcome measurement types: maturation, patency, access flow, inside diameter, occlusion rates of AVF.

5. No restrictions on publication types or languages.

## The exclusion criteria were

1. Non-RCTs and case reports were excluded.

2. Study on patients cross-treated with peritoneal dialysis.

3. No full texts available such as conference paper which without research details.

## Data extraction and quality assessment

Two evaluators (C.F.W. and P.J.H) extracted data from the included studies and assessed the quality studies by using the Cochrane Risk of Bias tool independently. If there was any disagreement in terms of the literature inclusion, the two evaluators would discuss with the third author (T.P.Y) to reach the consensus.

The evaluation items of the quality assessment including: 1) Selection bias: Are the randomization or blinding processes and methods described clearly? 2) Detection bias: Is the preventions of knowledge related to intervention allocations appropriately? 3) Lost data bias: Has missing data been addressed? 4) Reporting bias and 5) other biases [18].

## Statistical analysis

Statistical analysis of comparable data was performed by using Review Manager 5.4 software (Table 1). The result of continuous variable were analysis in standardized mean differences (SMD). Dichotomous variable results were revealed in risk ratios (RR) and 95% confidence intervals (CI). $\chi^2$ and $I^2$ tests ($I^2 > 50\%$ was considered significant heterogeneity) were used to assess data heterogeneity. When heterogeneity was low, fixed-effects models were applied to conduct the meta-analysis. Otherwise, a random effects model was used to reduce the effect of statistical heterogeneity [19]. The overall effects were examined by z-test, and $p$-values $< 0.05$ were considered as statistically significant. Besides, if there is any parameter with high heterogeneity, we use power analysis through R language (version 4.3.2 for Windows, 79 megabytes, 64 bit) to evaluate the power of evidence.

## Results

### Study selection

In Fig 1, 387 studies were identified after searching from multiple electronic databases, studies which irrelevant and duplicative were excluded. The full texts of the remaining 37 articles were

**Table 1. Study outcomes comparing FIR group and control group.**

| Outcomes | No. of studies | Sample size | | Heterogeneity(Total) | | | | SMD/RR(95%CI) | P value(Total) |
|---|---|---|---|---|---|---|---|---|---|
| | | FIR | Control | Chi$^2$ | Df | I$^2$% | P value | | |
| AVF malformation | 3 | 182 | 186 | 0.05 | 2 | 0 | 0.97 | 0.42 [0.28, 0.65] | P < 0.001 |
| Intervention for AVF | 2 | 132 | 135 | 0.15 | 1 | 0 | 0.70 | 0.49 [0.25, 0.96] | P = 0.04 |
| AVF occlusion within 12mo | 2 | 110 | 113 | 0.30 | 1 | 0 | 0.59 | 0.24 [0.08, 0.68] | P = 0.007 |
| Unassisted patency of AVF at 12month | 2 | 110 | 113 | 0.06 | 1 | 0 | 0.81 | 1.27 [1.09, 1.47] | P = 0.002 |
| Physiologic maturation of AVF at 3month | 2 | 110 | 113 | 0.32 | 1 | 0 | 0.57 | 1.22 [1.07, 1.39] | P = 0.002 |
| Clinical maturation of AVF within 12mo | 2 | 110 | 113 | 0.02 | 1 | 0 | 0.90 | 1.35 [1.14, 1.60] | P < 0.001 |
| Assessment of Qa0 | 2 | 110 | 113 | 0.04 | 1 | 0 | 0.85 | -0.01 [-0.28, 0.25] | p = 0.91 |
| Assessment of Qa1 | 4 | 224 | 233 | 9.54 | 3 | 69% | 0.02 | 0.43 [0.09, 0.76] | P = 0.01 |
| Assessment of Qa2 | 2 | 123 | 126 | 6.47 | 1 | 85 | 0.01 | 0.26 [-0.38, 0.90] | P = 0.42 |
| Assessment of Qa3 | 4 | 224 | 233 | 4.51 | 3 | 34 | 0.21 | 0.49 [0.26, 0.72] | P < 0.001 |
| Assessment of Qa12 | 2 | 110 | 113 | 40.06 | 1 | 98 | < 0.01 | 1.74 [-0.37, 3.84] | p = 0.11 |

CI = confidence interval, SMD = standard mean difference, RR = risk ratio.

further reviewed, and after filtering by inclusion and exclusion criteria, four eligible studies were included in the meta-analysis [7, 20–22]. During the literatures screening process, although both the papers of Lai et al. 2013 [23] and Lin et al. 2013 [15] used FIR therapy and targeted in patients with AVF; however, direct effects (such as patency and occlusion) of FIR on AVF were not indicated in Lin 2013 [15], and the participants in Lai et al. 2013 [23] included post AVF Percutaneous Transluminal Angioplasty (PTA) which may impact the statistical results hugely. Therefore, these two related RCTs were excluded for meta-analysis.

## Characteristics of included studies

In Table 2, the summary of the four literatures including: countries, study period, study designs, interventions, and sample sizes are represented.

Table 3 presents baseline characteristics of included studies, such as patient source, mean age, treatment, HD machines, protocol settings and frequency of FIR therapy.

The inclusion and exclusion criteria of the participant selection, the definitions of AVF malfunction in each study are presented in Table 4. The main model WS TY101 (WS Far Infrared Medical Technology Co., Ltd., Taipei, Taiwan) was used to provide the FIR therapy, which may produce wavelengths in the range between 3 to 25 μm (with peaks of 5 to 8 μm). The top radiator was placed at a height of 25–30 cm above the AVF surface and the therapy time is set for 40 minutes and 2–3 times per week.

## Risk of bias in included studies

Overall, only one trail did not reveal the randomization method, and three trials clearly described the randomization and allocation methods [7, 20–22]. Although all participants were not blinded, the assessors in four trials were blinded [7, 20–22]. Three studies reported participant dropouts [7, 20, 21]. All studies presented outcomes measurement methods and adverse events (Fig 2).

## Effects of FIR on AVF maturation and unassisted patency

In terms of fistula maturity, both physiological maturity at 3 months (RR = 1.22; 95%CI = 1.07 to 1.39; p = .002; Fig 3) and clinical maturity at 12 months (RR = 1.35; 95%CI = 1.14 to 1.60; p < .001; Fig 4) shows significant differences between the two groups, the FIR therapy groups were better than the control groups. The FIR groups also showed higher and significant

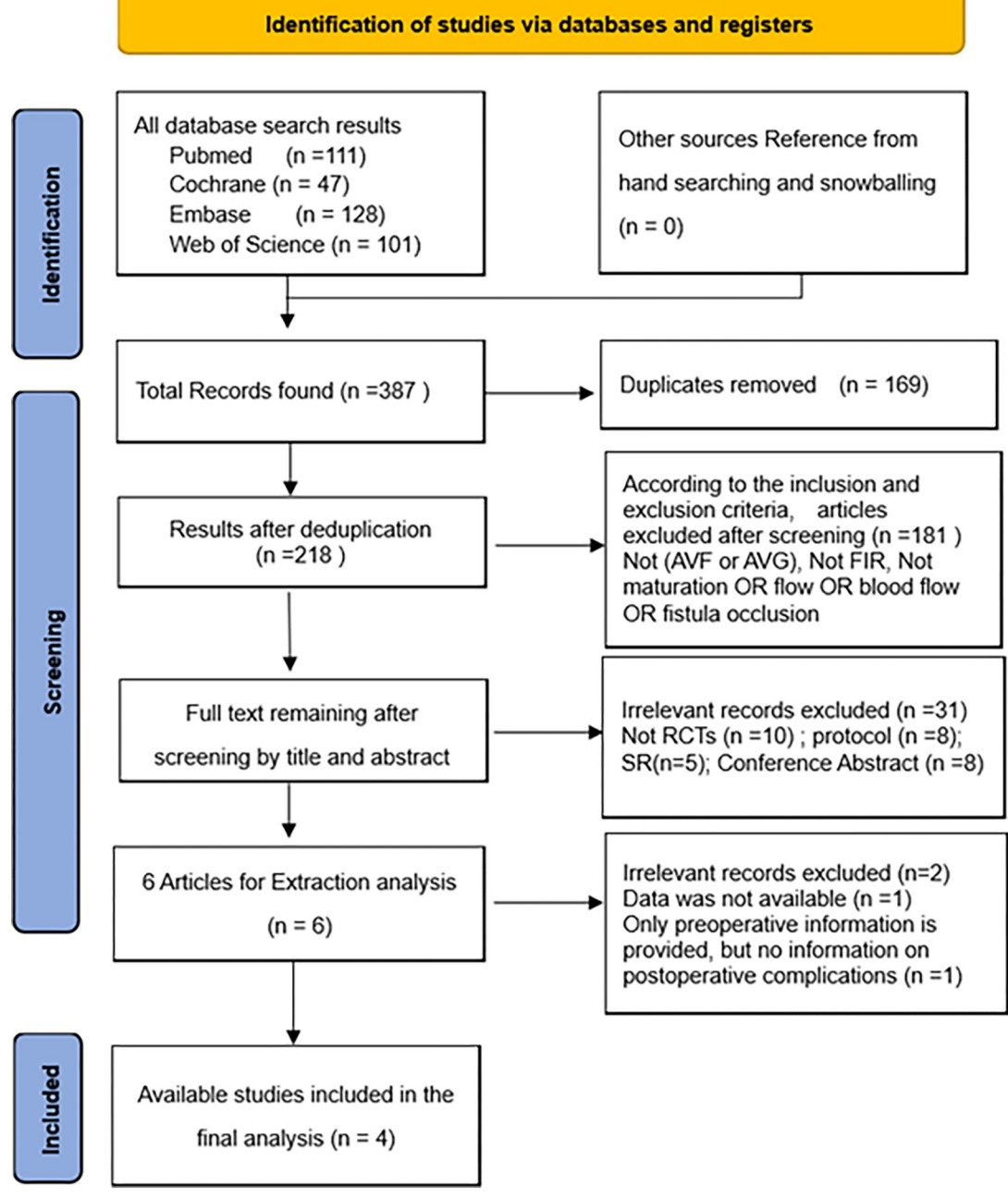

**Fig 1. PRISMA 2020 flow diagram.**

differences in unassisted patency (RR = 1.27; 95%CI = 1.09 to 1.47; $p$ = .002; $I^2$ = 0%; Fig 5) at 12 months without any intervention. There was no evidence of statistical heterogeneity ($df$ = 1 ($p$ = .57); $I^2$ = 0%; Fig 3; $df$ = 1($p$ = .90); $I^2$ = 0%; Fig 4; $df$ = 1($p$ = .81); $I^2$ = 0%; Fig 5). This suggests that FIR therapy is benefit in increasing maturation and patency of the AVF.

## Effects of FIR on access flow at different stages

In terms of the effects of FIR on Qa (access flow), there was no difference between the two groups when measured the Qa0 on baseline when AVF was established (RR = -0.01; 95%CI =

**Table 2. Summary of comparative studies included in meta-analysis.**

| Study | Country | Study Period | Study design | LE | Intervention | | Sample size | |
|---|---|---|---|---|---|---|---|---|
| | | | | | Trial | Control | Trial | Control |
| Lin 2007 | Taiwan | -2005.12.31 | RCT | IIa | FIR | Non-FIR | 72 | 73 |
| Lin 2013 | Taiwan | Not Provided | RCT | IIa | FIR | Non-FIR | 60 | 62 |
| An 2020 | India | 2020.08–2020.10 | RCT | IIa | FIR | Non-FIR | 51 | 56 |
| Chen 2022 | Taiwan | 2008.11–2010.08 | RCT | IIa | FIR | Non-FIR | 50 | 51 |

LE = Level of evidence; RCT = Randomized controlled trial; FIR = Far infrared therapy

**Table 3. Baseline characteristics of included studies.**

| Study | Patient source | Mean age (years) | Treatments | Machine | above the AVF(CM) | Protocol setting | Frequency of FIR |
|---|---|---|---|---|---|---|---|
| Lin 2007 | HD(AVF) | I:61.9±14.4 C:59.2 ± 15.0 | FIR | TY101 | 25 | 40min during HD | (3/week) |
| Lin 2013 | CKD(AVF) | 63.2 ± 18.5 | FIR | TY101N | 25 | 40min at Hospital or home | (3/week) |
| An 2020 | CKD(AVF) | I: 44.5±12.3 C: 47.0±14.2 | FIR | KS 9800 | 30 | 40min at Hospital or home | (2/week) |
| Chen 2022 | DKD(AVF) | I: 62.4±18.9 C: 62.3±14.2 | FIR | TY101 | 25 | 40min at Hospital or home | (3/week) |

**Table 4. Inclusion & exclusion criteria and definition of AVF malfunction for included studies.**

| Authors | Inclusion Criteria | Exclusion Criteria | Definition of AVF malfunction |
|---|---|---|---|
| Chen 2022 | 1) DKD with a GFR (eGFR) of 5–20 ml/min/1.73 m2, 2) AVF is expected to be established, 3) no dialysis or kidney transplantation within 3 months of study participation | 1) Patients receiving arterial grafts or tunneled HD catheters as permanent vascular access, 2) Patients who do not wish to receive HD, 3) Patients with grade III or IV heart failure, 4) Patients who have had a cardiovascular or cerebrovascular event within 3 months prior to screening or who are receiving interventional therapy, 5) Patients with advanced cancer with a life expectancy of less than 1 year. | Hematologic AVF with inadequate blood flow in patients who have not yet received HD or who require any intervention (angioplasty or surgery) to maintain AVF patency in patients who have received HD. |
| An 2020 | 1) CKD patients with GFR (eGFR) <30 ml/min/1.73 m2, 2) expected to create AVF, 3) age >13 years and <80 years. | 1) Complex vascular access surgery: e.g. re-establishment of AVF/prosthetic graft, 2) Patients with infection/sepsis within the last 2 weeks or history of antibiotics for more than 1 week, (3) Patients with thrombotic tendency, 4) History of macrovascular thrombosis, 5) Cardiovascular dysfunction, 6) Significant anemia Hb <6 gm %), 7) History of hypothyroidism. | None, only defined maturity |
| Lin 2013 | 1) CKD patients with GFR (eGFR) 5–20 ml/min/1.73 m2, 2) expected to have AVF, 3) age 18–80 years; and 4) not expected to receive dialysis or kidney transplantation within 3 years. | 1) patients receiving an arterial graft or tunneled HD catheter as permanent vascular access, 2) with Class III or IV heart failure, and 3) a cardiovascular or cerebrovascular event within 3 months prior to screening or receiving interventional therapy to receive interventional therapy. | 1) AVF without tremor thrombosis in patients not yet receiving HD, 2) any type of AVF intervention (surgery or angioplasty) as a result of patients receiving HD. |
| Lin 2007 | 1) hospitalized for at least 6 months for 3 weekly 4-hour HD sessions, 2) AVF has been used as vascular access for more than 6 months without intervention in the last 3 months, 3) AVF was created by a hospitalized cardiovascular surgeon during a standardized surgical procedure | Not presented | AVF requires any intervention (surgery or angioplasty) to correct the occlusion or dysfunction, but will exclude failure to maintain 200 mL/min of extracorporeal flow during HD after the following non-stenosis related events: infectious complications. |

DKD = diabetic kidney disease; GFR = glomerular filtration rate; CKD = chronic kidney disease

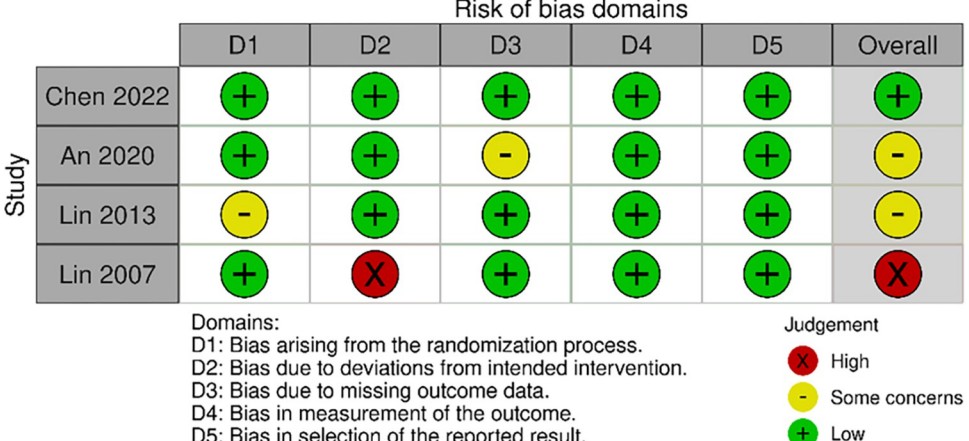

**Fig 2. Risk of bias in included studies.**

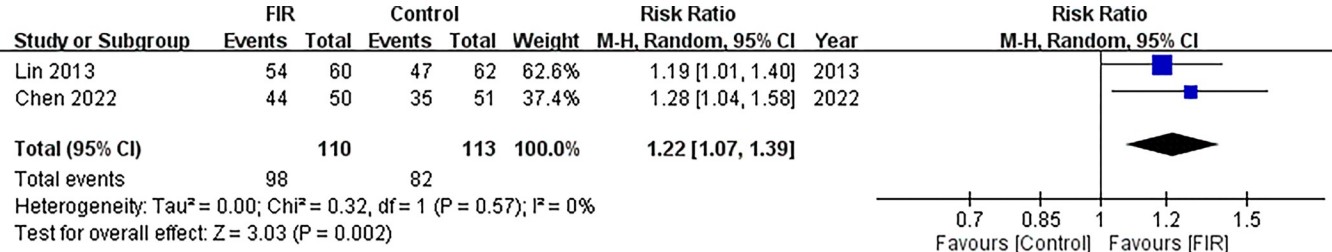

**Fig 3. Forest plot of comparison.** 1 Far infra-Red phototherapy for AVF malformation, outcome: 1.5 Physiologic maturation of AVF at 3mo.

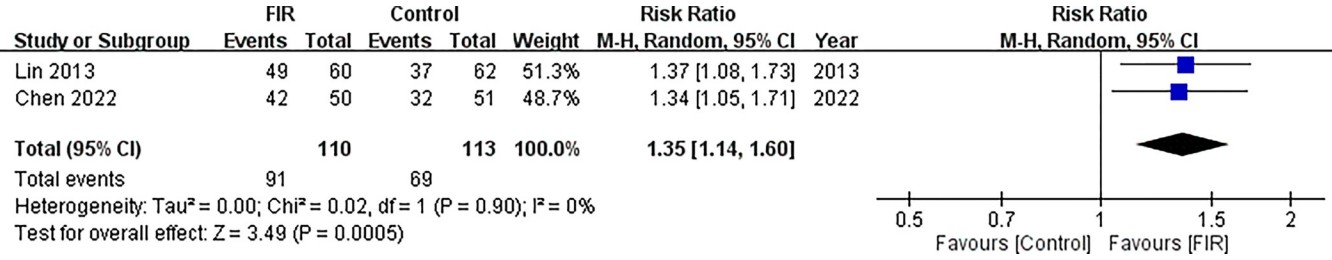

**Fig 4. Forest plot of comparison.** 1 Far infra-Red phototherapy for AVF malformation, outcome: 1.6 Clinical maturation of AVF within 12mo.

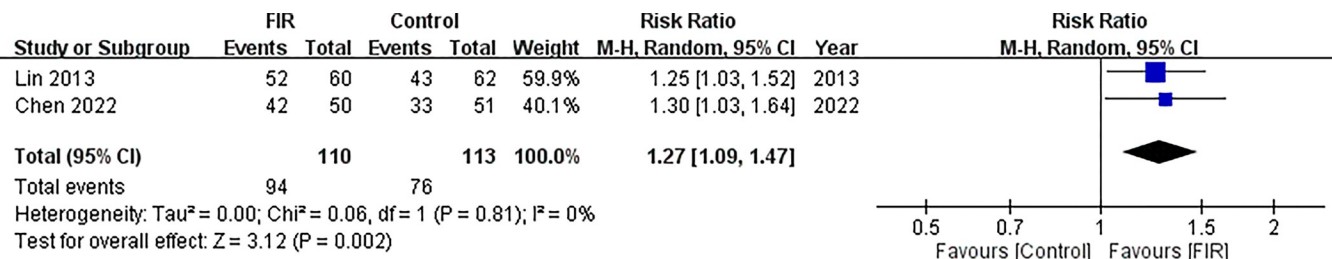

**Fig 5. Forest plot of comparison.** 1 Far infra-Red phototherapy for AVF malformation, outcome: 1.4 Unassisted patency of AVF at 12 mo.

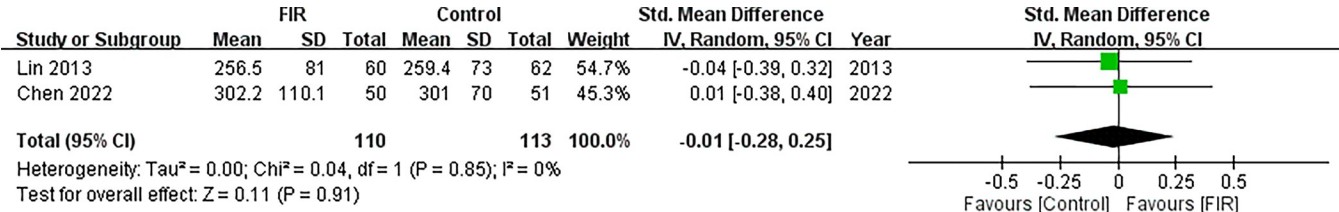

**Fig 6. Forest plot of comparison.** 1 Far infra-Red phototherapy for AVF malformation, outcome: 1.7 Assessment of Qa0 (mL/min).

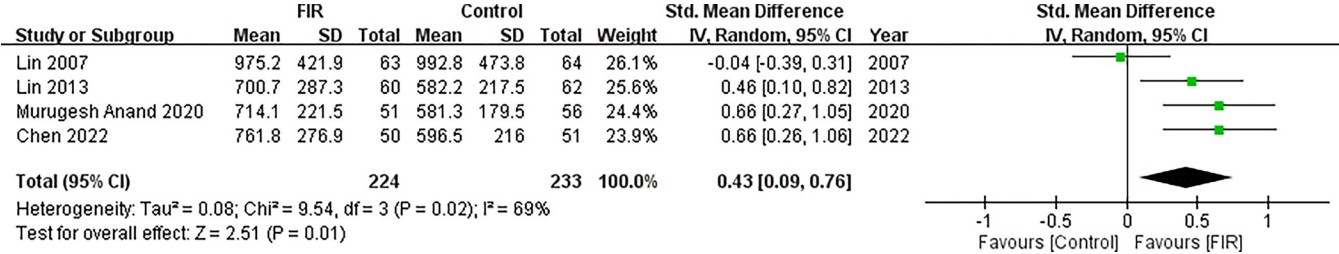

**Fig 7. Forest plot of comparison.** 1 Far infra-Red phototherapy for AVF malformation, outcome: 1.8 Assessment of Qa1 (mL/min).

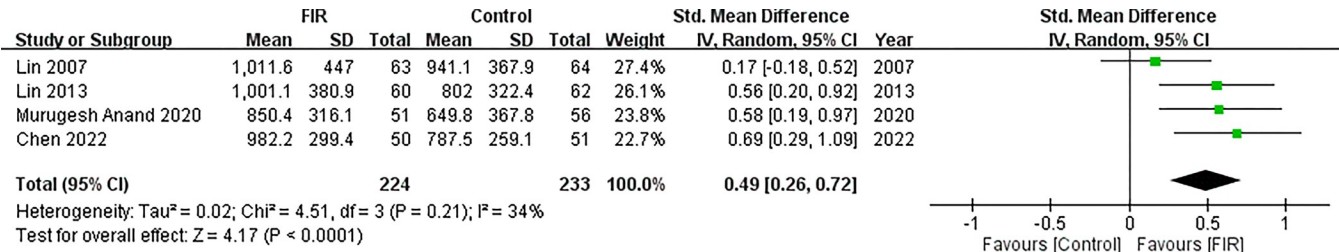

**Fig 8. Forest plot of comparison.** 1 Far infra-Red phototherapy for AVF malformation, outcome: 1.10 Assessment of Qa3 (mL/min).

-0.28 to 0.25; $p$ = .91; Fig 6). All studies had Qa1 in the 1st-month and Qa3 in the 6th-months, Qa2 had only done in the 3rd-months in Lin et al. [20] and Lin et al. [21], and Qa4 had been completed in the 12th-months in Chen et al. [7] and Lin et al. [20]. In FIR therapy groups, all studies showed the increasing of vascular access flow (RR = 0.43; 95%CI = 0.09 to 0.76; $p$ = .01; 1-month; Fig 7; RR = 0.49; 95%CI = 0.26 to 0.72; $p$ < .001; 6-months; Fig 8). There was no statistical difference in Qa2 at 3rd-months (RR = 0.26; 95%CI = -0.38 to 0.90; $p$ = .42; Fig 9) and 12th-months (RR = 1.74; 95%CI = -0.37 to 3.84; $p$ = .11, Fig 10) after AVF establishment. The four studies included in the meta-analysis showed considerable heterogeneity at 1st-month ($df$ = 3 ($p$ = .02); $I^2$ = 69%; Fig 7), 3-months ($df$ = 1 ($p$ = .01); $I^2$ = 85%; Fig 9) and 12th-months ($df$ = 1 (p < .01); $I^2$ = 98%; Fig 10) and a low heterogeneity at 6th-months ($df$ = 3 ($p$ = .21); $I^2$ = 34%; Fig 8) in each sub-group. Therefore, the effects of FIR on vascular access flow needs to be further discussed and investigated. Besides, high heterogeneity had been detected in Qa1 ($I^2$ = 69%), Qa2 ($I^2$ = 85%), and Qa4 ($I^2$ = 98%). The result of power analysis showed statistical power on each parameter as Qa1 (Power = 89.42%), Qa2 (Power = 30.44%), and Qa4 (Power = 100%).

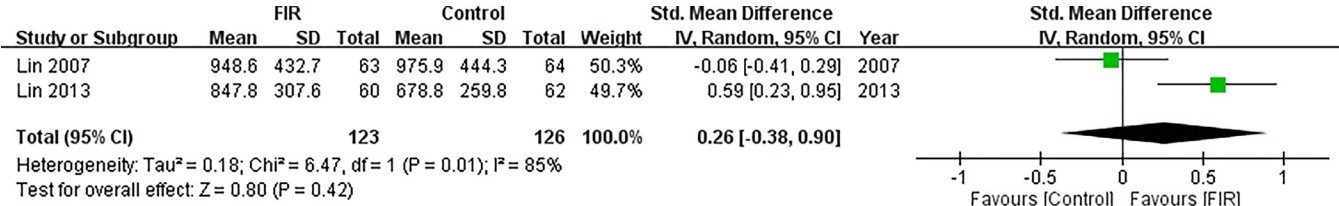

**Fig 9. Forest plot of comparison.** 1 Far infra-Red phototherapy for AVF malformation, outcome: 1.9 Assessment of Qa2 (mL/min).

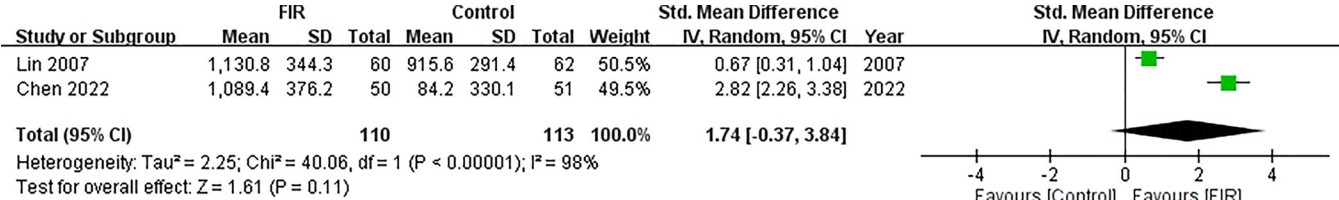

**Fig 10. Forest plot of comparison.** 1 Far infra-Red phototherapy for AVF malformation, outcome: 1.11 Assessment of Qa4 (mL/min).

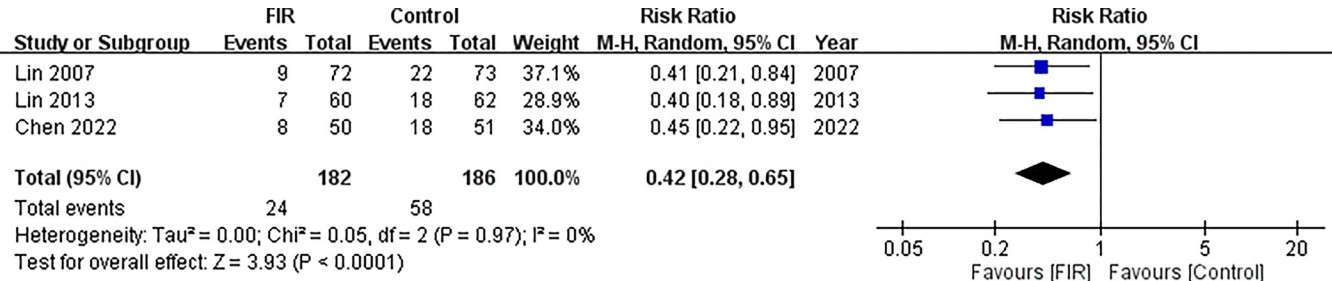

**Fig 11. Forest plot of comparison.** 1 Far infra-Red phototherapy for AVF malfunction, outcome: 1.1 AVF malfunction.

### Effect of FIR on malfunction within 12-months

Three studies analyzed AVF malfunction status by using a random effects model (n = 368). The meta-analysis results revealed a lower failure rate of AVF in the FIR intervention groups (RR = 0.42; 95% CI = 0.28 to 0.65; $p < .001$; Fig 11) and no heterogeneity was found ($df = 2(p = .970)$; $I^2 = 0\%$).

### Effect of FIR on intervention and occlusion rate within 12-months

In the FIR group, further interventions such as PTA or thrombectomy to maintain AVF function was lower (RR = 0.49; 95% CI = 0.025 to 0.96; $p = .04$; Fig 12) and the obstruction rates within 12-months was much lower in the FIR groups (RR = 0.24; 95% CI = 0.08 to 0.68; $p = .007$, Fig 13) and no statistical heterogeneity ($df = 1$ ($p = .70$); $I^2 = 0\%$; Fig 12; $df = 1$ ($p = .59$); $I^2 = 0\%$; Fig 13). This suggested advantages of FIR therapy in reducing AVF occlusion incidences.

## Discussion

This meta-analysis integrated data from four RCTs, there were no significant differences in the participants' demographic characteristics. The meta-analysis results supported that FIR

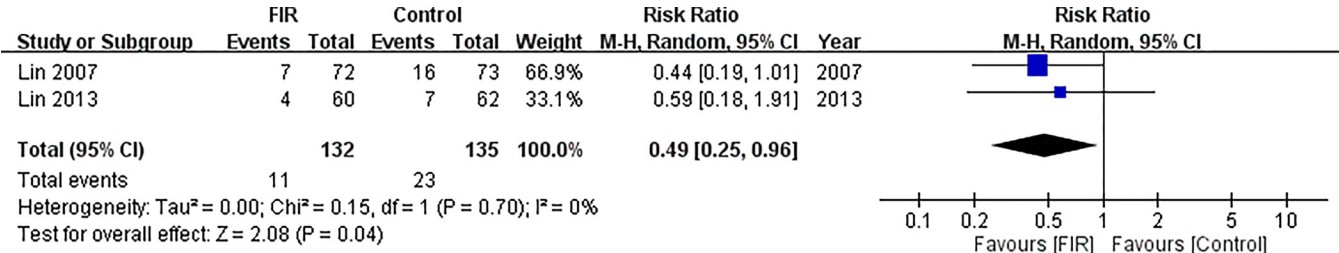

**Fig 12. Forest plot of comparison.** 1 Far infra-Red phototherapy for AVF malformation, outcome: 1.2 Intervention for AVF.

therapy improves AVF patency, reduces obstructions, and increases fistula maturity; which is consistent with the research findings of Bashar et al. 2014 [16] and Wan et al. 2017 [12].

In the effects of FIR on access flow analysis, all studies showed that the access flow in the FIR group was better than the control group and reached a statistically significant difference. However, in this meta-analysis, the results showed only in the control group at Qa1 and Qa3 revealed statistically significant effects; while no significant differences were found at Qa2 and Qa4 (Table 1). These results were different from the research of Wan et al. [12] are in 2017, and this may be related to the small number of articles included the Qa2 and Qa4 for analyses. In addition, the study of Lin et al. [20] in 2007 recruited the patients who had already undergone HD with AVF and not with newly established AVF. Therefore, the effects of FIR on access flow need to be further investigated.

The fistula maturation rates at the 12th month after AVF established in the CKD patients was 58%, and 47.5% patients with newly established AVF required further interventions [5]. This has echoed the percentage of maturation at the 12th month after AVF establishment in the control group in this study (60%-62.7%) [7, 21]. The maturation rate of newly established AVF in FIR therapy groups at 12th- month was 82%-84% in this meta-analysis, which was similar to the Braun and Khayat's (2021) study. This result may indicate that FIR could improve the maturation rate of new AVF.

Previous study has indicated that the 12th-month unassisted patency after AVF established was 51–72% [4]; but in this meta-analysis, patients with FIR had a patency of 84%-87% with no intervention [7, 21]. The 12th-month unassisted patency in FIR therapy groups in this meta-analysis was similar to the AVF patency (87%) which requiring endovascular or surgical procedures [5]. The cost of performing an endovascular or surgical procedure ranges from $2,000-$4,000 [24], a far-infrared machine (WS TY101 model, for example) costs only about $1,500. Therefore, improving AVF patency and increasing fistula maturity through FIR treatment might significantly reduce the medical costs, which is a 'win-win' situation for both patients and medical care units.

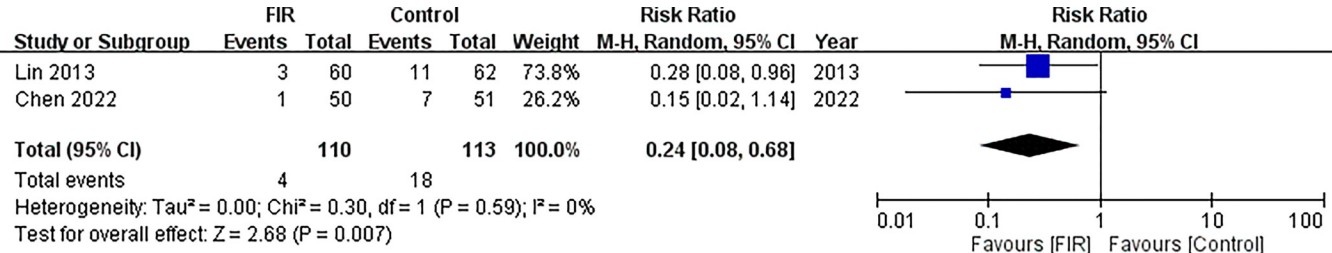

**Fig 13. Forest plot of comparison.** 1 Far infra-Red phototherapy for AVF malformation, outcome: 1.3 AVF occlusion within 12mo.

The effects of FIR therapy on AVF maturation may result from: (1) inhibition of intimal hyperplasia; (2) reduction of oxidative stress; (3) inhibition of inflammation; and (4) improvement of endothelial function. Studies had demonstrated that FIR may reduce asymmetric dimethylarginine (ADMA), as a result the inflammation and endothelial hyperplasia could be reduced; consequently, the AVF maturation and patency could be enhanced [7, 10, 14, 15, 25, 26]. The effect of FIR treatment on access flow may due to the thermal effects, which would induce vasodilatation and increase access flow; skin surface temperature may only raise to 38–39˚C which may avoid burns caused by hot compress [12, 13].

Among the included literature, 2013 Lin et al. [21] and 2020 An et al. [22] both mentioned the effect of FIR on AVF diameter, but since 2020 An [22] included AVF with radio-cephalic fistulas(RCF) and brachio-cephalic fistula (BCF), but 2013 Lin [21] included only RCF, this paper did not compare the effect of FIR on AVF diameter and could not confirm whether FIR Therefore, this paper does not compare the effect of FIR on AVF diameter, and cannot confirm whether FIR has an effect on AVF diameter. The effect of FIR on cardiac output, blood pressure, and total peripheral resistance was only mentioned in the study by 2007 Lin et al. [20], so no analysis of the relevant data was conducted.

There are some study limitations in this meta-analysis. Only four studies were included, which represented a slightly insufficient sample size. Among the four selected studies, three studies had 12 months intervention time, but one study only had a 4-week intervention time; which was inconsistent in intervention duration. Only the AVF of RCF was compared, so that the effect of FIR on BCF could not be investigated. Heterogeneity was persistent in the access flow analysis, although subgroups were used for analysis and random effects model was applied; the heterogeneity may still affect the statistical results. The longest FIR irradiation lasted about 12 months, how the FIR affects primary AVF patency and obstruction rates in more than 12 months remains unknown. Another irradiation duration issue is that whether only 3 months or 6 months intervention could achieve the effect of increasing AVF maturation and reducing subsequent AVF obstruction rates or not? Among the four studies, three were conducted in Taiwan and one is in India, which may have some limitations when the meta-analysis results would like to extrapolate to other populations like European and American countries.

## Conclusion

In summary, FIR therapy could effectively improve access flow whether the AVF is new or not; and for new AVFs, FIR therapy can increase AVF maturation and reduce obstruction. Therefore, FIR therapy is a non-invasive treatment modality to improve AVF maturation and patency. FIR therapy is convenient for patients' use. FIR therapy is also recommended for patients who have DM nephropathy with poor vascular conditions, because it can be used prophylactically to increase the success rate of fistula establishment and reduce the harm caused by poor AVF maturation or obstruction.

## Declaration

### Protocol registration number

Inplasy, the International Platform of Registered Systematic Review and Meta-analysis Protocols, had approved this study with certification number INPLASY202340020 (Published on 07 April, 2023).

## Supporting information

**S1 Checklist.**
(DOCX)

**S1 File.**
(DOCX)

## Acknowledgments

The authors would like to thank all their colleagues and students who contributed to this study. Special thanks to Chia-Lung Shih for his great efforts in suggestion of statistical analysis.

## Author Contributions

**Conceptualization:** Chiu-Feng Wu, Po-Hsiang Huang.

**Data curation:** Chiu-Feng Wu, Pin-Jui Huang.

**Formal analysis:** Pin-Jui Huang.

**Funding acquisition:** Pin-Jui Huang.

**Investigation:** Chiu-Feng Wu, Tzu-Chen Lin, Pin-Jui Huang.

**Methodology:** Chiu-Feng Wu, Tzu-Pei Yeh, Tzu-Chen Lin, Pin-Jui Huang.

**Project administration:** Chiu-Feng Wu, Tzu-Pei Yeh, Pin-Jui Huang.

**Resources:** Chiu-Feng Wu, Tzu-Pei Yeh, Pin-Jui Huang.

**Software:** Chiu-Feng Wu, Tzu-Pei Yeh, Pin-Jui Huang.

**Supervision:** Chiu-Feng Wu, Tzu-Pei Yeh, Po-Hsiang Huang.

**Validation:** Po-Hsiang Huang.

**Visualization:** Pin-Jui Huang.

**Writing – original draft:** Chiu-Feng Wu, Pin-Jui Huang.

**Writing – review & editing:** Chiu-Feng Wu, Pin-Jui Huang.

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
