## [Decision Letter · Decision Letter 0]

28 Nov 2023

PONE-D-23-21091Effects of Far Infrared Therapy in Hemodialysis Arterio-Venous Fistula Maturation: A Meta-analysisPLOS ONE

Dear Dr. Huang,

Thank you for submitting your manuscript to PLOS ONE. After careful consideration, we feel that it has merit but does not fully meet PLOS ONE’s publication criteria as it currently stands. Therefore, we invite you to submit a revised version of the manuscript that addresses the points raised during the review process.

We look forward to receiving your revised manuscript.

Kind regards,

Ahmet Murt

Academic Editor

PLOS ONE

Journal Requirements:

Additional Editor Comments:

In this meta-analysis, authors analyzed possible affect of FIR therapy on AVF maturation. In brief they found that FIR therapy may improve AVF potency. The topic may be of interest to the nephrology readership.

However there are metrhological pitfalls that needs to be corrected.

For example this important study was not included in the analysis and it's even not mentioned as a reference: https://www.ncbi.nlm.nih.gov/pmc/articles/PMC8379732/.

I would like to invite authors to re-check their search strategy and re-structure their paper.

Reviewers' comments:

Reviewer's Responses to Questions

**Comments to the Author**

1. Is the manuscript technically sound, and do the data support the conclusions?

Reviewer #1: Yes

Reviewer #2: Yes

Reviewer #3: Yes

2. Has the statistical analysis been performed appropriately and rigorously? 

Reviewer #1: Yes

Reviewer #2: No

Reviewer #3: Yes

3. Have the authors made all data underlying the findings in their manuscript fully available?

Reviewer #1: Yes

Reviewer #2: Yes

Reviewer #3: Yes

4. Is the manuscript presented in an intelligible fashion and written in standard English?

Reviewer #1: Yes

Reviewer #2: Yes

Reviewer #3: Yes

5. Review Comments to the Author

Reviewer #1: To the authors: I commend you on the quality of your writing. Although I have some reservations about a four-article review, you have done an excellent job with its composition. I believe those interested in infrared technology for AVFs will find this article to be a pleasant read. Best regards,

Reviewer #2: Dear Authors

I read your manuscript with great interest. Overall idea is good and paper is well established. However I have some major concerns as follows:

Meta-analysis should be conducted when a group of studies is sufficiently homogeneous in terms of subjects involved, interventions, and outcomes to provide a meaningful summary. However, it is often appropriate to take a broader perspective in a meta-analysis than in a single clinical trial.

The reliability of the conclusion is influenced by factors like the presence of bias, heterogeneity among the studies (PICO), and the overall certainty of the evidence. If there is minimal bias and high certainty in the evidence, the results from these two studies should have the potential to provide a meaningful conclusion.

The power analyse should be made

Reviewer #3: I congratulate the authors for their efforts. They conducted a successful systematic review and meta-analysis evaluating the effects of FIR treatment on fistula maturity and patency, including four RCTs. I think that the design, methodology and ethical requirements of the study were appropriate and well written. Although the shortcomings of two similar meta-analyses were mentioned in the introduction, I think that this study does not show very different and strong results from the previous two meta-analyses. Please discuss any real strengths and differences from the other two meta-analyses by highlighting them in the discussion section.

In addition:

- On page 10 line 94, in the sentence (2. Participant Type: ESRD patients diagnosed with CKD and AVF,...), AVF is not a diagnosis but a vascular access route for HD in CKD patients. Please edit this sentence.

6. PLOS authors have the option to publish the peer review history of their article (what does this mean?). If published, this will include your full peer review and any attached files.

Reviewer #1: No

Reviewer #2: No

Reviewer #3: No

---

## [Author Response · Author response to Decision Letter 0]

22 Feb 2024

Responses to Reviewer 1:

1.Comment: To the authors: I commend you on the quality of your writing. Although I have some reservations about a four-article review, you have done an excellent job with its composition. I believe those interested in infrared technology for AVFs will find this article to be a pleasant read. Best regards,

Response: We thank the reviewer for giving recognition to this article. Indeed, four-article review had its limitation, such as insufficient sample size and statistically power. We did our best to lower the bias and increase the power of evidence by using random model, standardized data assessment with standardized mean difference, and subgroup analysis. The above information is all included in the section on limitations. We thank the reviewer again for his affirmation of this article.

 

Responses to Reviewer 2:

1.Comment: I read your manuscript with great interest. Overall idea is good and paper is well established. However I have some major concerns as follows: Meta-analysis should be conducted when a group of studies is sufficiently homogeneous in terms of subjects involved, interventions, and outcomes to provide a meaningful summary. However, it is often appropriate to take a broader perspective in a meta-analysis than in a single clinical trial. The reliability of the conclusion is influenced by factors like the presence of bias, heterogeneity among the studies (PICO), and the overall certainty of the evidence. If there is minimal bias and high certainty in the evidence, the results from these two studies should have the potential to provide a meaningful conclusion. The power analysis should be made.

Response: We thank the reviewer for pointing out that power of evidence should be carefully evaluated. Throughout all the results, there were three parameters with high heterogeneity, including assessment flow Qa1, Qa2, and Qa4. We use the power analysis through R language (version 4.3.2 for Windows, 79 megabytes, 64 bit) for evaluate the power of evidence. The result of power analysis was listed in the revised manuscript. 

 

Responses to Reviewer 3:

1.Comment: I congratulate the authors for their efforts. They conducted a successful systematic review and meta-analysis evaluating the effects of FIR treatment on fistula maturity and patency, including four RCTs. I think that the design, methodology and ethical requirements of the study were appropriate and well written. Although the shortcomings of two similar meta-analyses were mentioned in the introduction, I think that this study does not show very different and strong results from the previous two meta-analyses. Please discuss any real strengths and differences from the other two meta-analyses by highlighting them in the discussion section.

In addition: - On page 10 line 94, in the sentence (2. Participant Type: ESRD patients diagnosed with CKD and AVF,...), AVF is not a diagnosis but a vascular access route for HD in CKD patients. Please edit this sentence.

Response: We thank the reviewer for pointing out that the strength and difference from the other two meta-analysis were not clear yet. Nevertheless, although those key parameters presented similar outcome, we have mentioned new data in our article, such as access flow at 1,3 and 6 months (Qa1, Qa2, Qa3), which related to the clinical maturation of arterio-venous fistula. Besides, those key parameters were separated in the two meta-analyses (Bashar et al. mentioned AVF patency and surgical intervention for AVF malfunction; Wan et al. mentioned assessment flow < six months or > six months, AVF diameter, AVF patency, AVF occlusion, needling pain), it was hard to figure out whether all of the parameters were statistically significant or not. In our article, we integrated all the parameters mentioned before, and provided our answer in the article.

And for the part of addition, we thank the reviewer for pointing out these errors. The reviewer is correct, and we apologize for the inappropriate presentation. We have now deleted the misunderstanding words and change the sentence to " 2. Participant Type: patients diagnosed with CKD or ESRD, and receiving regular HD treatment with AVF "

---

## [Decision Letter · Decision Letter 1]

16 May 2024

PONE-D-23-21091R1Effects of Far Infrared Therapy in Hemodialysis Arterio-Venous Fistula Maturation: A Meta-analysisPLOS ONE

Dear Dr.Huang,

Thank you for submitting your manuscript to PLOS ONE. After careful consideration, we feel that it has merit but does not fully meet PLOS ONE’s publication criteria as it currently stands. Therefore, we invite you to submit a revised version of the manuscript that addresses the points raised during the review process.

. 

We look forward to receiving your revised manuscript.

Kind regards,

Ahmet Murt

Academic Editor

PLOS ONE

Journal Requirements:

Additional Editor Comments:

In this revised version of the manuscript, our impartial reviewers are generally satisfied with your explanations. However I see that there are some typo and grammar errors. Please correct them (a native speaker helper is recommended). One example: Previous have indicated that the 12th-month unassisted patency after AVF established was 51-72%. Previous studies? or Previously it was?

Please check the whole manuscript to have a better language.

Reviewers' comments:

Reviewer's Responses to Questions

**Comments to the Author**

1. If the authors have adequately addressed your comments raised in a previous round of review and you feel that this manuscript is now acceptable for publication, you may indicate that here to bypass the “Comments to the Author” section, enter your conflict of interest statement in the “Confidential to Editor” section, and submit your "Accept" recommendation.

Reviewer #2: All comments have been addressed

Reviewer #3: All comments have been addressed

2. Is the manuscript technically sound, and do the data support the conclusions?

Reviewer #2: Yes

Reviewer #3: Yes

3. Has the statistical analysis been performed appropriately and rigorously? 

Reviewer #2: Yes

Reviewer #3: Yes

4. Have the authors made all data underlying the findings in their manuscript fully available?

Reviewer #2: Yes

Reviewer #3: Yes

5. Is the manuscript presented in an intelligible fashion and written in standard English?

Reviewer #2: Yes

Reviewer #3: Yes

6. Review Comments to the Author

Reviewer #2: Dear Authors

I reviewed the revised version of the manuscript

I congratulate to you for successful revision. The paper can be published as is

Reviewer #3: I would like to thank the authors for their responses to previous suggestions and for their efforts to improve the manuscript accordingly.

7. PLOS authors have the option to publish the peer review history of their article (what does this mean?). If published, this will include your full peer review and any attached files.

Reviewer #2: No

Reviewer #3: **Yes: **Eyüp Serhat Çalık

---

## [Author Response · Author response to Decision Letter 1]

29 Jun 2024

We've checked the grammer and spelling mistakes, and uploaded the revised version to the platform.

Thank you very much!

---

## [Editor Report · Decision Letter 2]

9 Jul 2024

Effects of Far Infrared Therapy in Hemodialysis Arterio-Venous Fistula Maturation: A Meta-analysis

PONE-D-23-21091R2

Dear Dr. Huang

We’re pleased to inform you that your manuscript has been judged scientifically suitable for publication and will be formally accepted for publication once it meets all outstanding technical requirements.

Kind regards,

Ahmet Murt

Academic Editor

PLOS ONE

Additional Editor Comments (optional):

I have no furrther comments.
---

## [Editor Report · Acceptance letter]

19 Jul 2024

PONE-D-23-21091R2 

PLOS ONE

Dear Dr. Huang, 

I'm pleased to inform you that your manuscript has been deemed suitable for publication in PLOS ONE. Congratulations! Your manuscript is now being handed over to our production team.

Kind regards, 

on behalf of

Dr. Ahmet Murt 

Academic Editor

PLOS ONE